# Prediction of ^57^Fe Mössbauer Nuclear Quadrupole Splittings with Hybrid and Double-Hybrid Density Functionals

**DOI:** 10.3390/ijms26062821

**Published:** 2025-03-20

**Authors:** Yihao Zhang, Haonan Tang, Wenli Zou

**Affiliations:** 1Institute of Modern Physics, Northwest University, Xi’an 710127, China; zhangyh1245@163.com (Y.Z.); tang123hn@163.com (H.T.); 2School of Physics, Northwest University, Xi’an 710127, China; 3Shaanxi Key Laboratory for Theoretical Physics Frontiers, Xi’an 710127, China

**Keywords:** Mössbauer spectroscopy, electric field gradient, strongly correlated systems, exact two-component relativistic Hamiltonian, copper monofluoride

## Abstract

As a crucial parameter in Mössbauer spectroscopy, nuclear quadrupole splitting (NQS) exhibits a strong dependence on quantum chemistry methods, which makes accurate theoretical predictions challenging. Meanwhile, the continuous emergence of new density functionals presents opportunities to advance current NQS research. In this study, we evaluate the performance of eleven hybrid density functionals and twelve double-hybrid density functionals, selected from widely used functionals and newly developed functionals, in predicting the NQS values of the ^57^Fe nuclide for 32 iron-containing molecules within about 70 atoms. The calculations have incorporated scalar relativistic effects using the exact two-component (X2C) Hamiltonian. In general, the double-hybrid functional PBE-0DH demonstrates superior performance compared to the experimental values, achieving a mean absolute error (MAE) of 0.20 mm/s. Meanwhile, *r*SCAN38 is the best hybrid functional for our database with an MAE = 0.25 mm/s, and it offers a significant advantage in computational efficiency over PBE-0DH. The +/− sign of NQS has also been considered in our error statistics when it has a clear physical meaning; if neglected, the errors of many functionals decrease, but PBE-0DH and *r*SCAN38 remain unaffected. Notably, when calculating ferrocene [Fe(C_5_H_5_)_2_], which involves strong static correlations, all hybrid functionals that incorporate more than 10% exact exchange fail, while several double-hybrid functionals continue to deliver reliable results. In addition, we encountered two particularly challenging species characterized by strong static correlations: [Fe(H_2_O)_5_NO]^2^+ and FeO_2_^−^-porphyrin. Unfortunately, none of the density functionals tested in our study yielded satisfactory results for the two cases since the density functional theory (DFT) is a single-determinant approach, and it is imperative to explore large-scale multi-configurational methods for these species. This research offers valuable guidance for selecting density functionals in Mössbauer NQS calculations and serves as a reference point for the future development of new density functionals.

## 1. Introduction

Any atomic nucleus with a nuclear spin quantum number (*I*) greater than 1/2 displays an ellipsoidal distribution of nuclear charge, leading to a quadrupole electric field surrounding the nucleus, referred to as the nuclear quadrupole moment (NQM). It has been found that more than 40% of stable atomic nuclei possess NQMs. NQM can interact with the electric field gradient (EFG) produced by surrounding charges, known as the nuclear quadrupole interaction (NQI). NQI can be investigated using a range of spectroscopic techniques, each yielding a distinct observable. For instance, in Mössbauer spectroscopy, the NQI is quantified through nuclear quadrupole splitting (NQS), while, in gas-phase microwave spectroscopy, NQI is related to the nuclear quadrupole coupling constant (NQCC).

In Mössbauer spectroscopy, three distinct types of hyperfine interactions can be identified [1]:The electric monopole interactions between protons in atomic nuclei and electrons (primarily *s*-electrons and, to a lesser extent, *p*- or $p_{1/2}$-electrons caused by relativistic effects) extend into the nuclear region, which can be measured through the isomer shifts ($\delta^{\mathrm{IS}}$).The NQM of a nucleus with I> 1/2 interacts with the EFG generated by asymmetric environmental charges (mainly composed of *p*- and *d*-electrons from the target atom as well as nuclei and electrons from neighboring atoms), leading to the aforementioned NQS.The interactions between the atomic nuclear magnetic dipole moment and the surrounding magnetic field contribute to the Mössbauer magnetic hyperfine Zeeman splitting.

Among these, two essential Mössbauer parameters, namely $\delta^{\mathrm{IS}}$ and NQS, can be theoretically examined using modern quantum chemical methods by analyzing the calculated contact density (or effective contact density) [2] and EFG [3], respectively. Of the more than 80 nuclides exhibiting the Mössbauer effects, the majority of theoretical investigations have concentrated on the ^57^Fe nuclide, which serves as a valuable indicator for deducing the bonding characteristics, valence states, and local spins of iron atoms within various compounds. In the literature, nearly all the theoretical studies on the Mössbauer spectroscopy of iron compounds have been performed using density functional theory (DFT), but only a limited number of recent studies among them have taken scalar relativistic effects into account (cf. the summaries in References [2,3]). Given the strong electronic correlations frequently arising in iron compounds, traditional density functionals often fall short in effectively addressing these systems. Consequently, it is essential to identify more appropriate density functionals for accurately calculating the Mössbauer parameters of iron compounds.

There have been many recent studies in the literature that evaluate density functionals (for example, see References [4,5,6,7,8,9,10,11,12,13,14,15]), but the conclusions drawn from these investigations often differ significantly, influenced by the specific properties being examined and the molecular systems under consideration. Notably, some newly developed hybrid functionals and double-hybrid functionals exhibit better accuracy than others [6,7,8,9,12,14]. In contrast, traditional functionals frequently inaccurately describe the electronic structures and properties of strongly correlated systems. In the most extreme cases, such as diatomic CuX (X = H, F, Cl, Br, and I) molecules, certain commonly used functionals can produce qualitative errors regarding the +/− signs of EFG for the copper nuclide [16,17,18,19,20]. Over the past decade, many new hybrid and double-hybrid density functionals have been developed, but, to our knowledge, there is currently no comprehensive database dedicated to the functional development that accounts for the electron distribution around the nucleus. Consequently, the accuracy of these functionals in calculating Mössbauer parameters, particularly for iron compounds, remains uncertain.

In a recent paper [3], we employed the spin-free X2C (exact two-component) [21,22,23,24] relativistic DFT method to compute the NQS values of ^57^Fe for seventeen iron compounds, each containing up to 60 atoms. We utilized two randomly selected density functionals: the hybrid functional B3LYP and the double-hybrid functional DSD-PBEP86. Our findings indicated that the latter functional yielded better results than the former. In this paper, we extend our investigation to include 32 iron compounds, incorporating a range of hybrid and double-hybrid functionals, with a particular emphasis on the latest developments in this area.

This paper is organized as follows: The results are presented in Section 2 and discussed in Section 3, respectively. Section 4 presents the fundamental formulas, defines relevant symbols, and outlines the computational details. Finally, some conclusions are drawn in the last section.

## 2. Results

### 2.1. Preliminary Evaluation of Density Functionals

Given the vast number of hybrid and double-hybrid functionals available for testing (see Section 4.2), a preliminary screening is essential. Small molecules containing iron often exhibit open-shell degenerate states with significant multi-configurational characteristics or lack reliable experimental data [25], and, therefore, we chose the closed-shell CuF molecule as our initial testing system, which was originally proposed by Bast and Schwerdtfeger for evaluating new functionals [17]. While CuF may appear straightforward for calculations at first glance, it presents considerable challenges when approached with conventional functionals [17,18,19,20], as well as with advanced ab initio methods [26]. Compounds containing 3*d* metals are widely recognized for their strong correlations, which make them particularly challenging to calculate [27]. Several explanations have been proposed to account for this phenomenon from the perspective of static–dynamic correlations [28], whereas the phenomenological explanation in condensed matter physics seems not helpful, especially given that DFT with a Hubbard U correction usually performs worse than hybrid DFT methods [29]. A prevalent explanation suggests that these strong correlations stem from the competition between the 3$d^{n}$ and 3$d^{n-1}$ occupations, along with the large discrepancies in their correlation errors caused by the different spatial distributions of the two types of 3*d* orbitals. Another explanation claims that significant 3d→4d local excitations contribute to the emergence of strong correlations. Anyhow, the strong correlations present in 3*d* metal systems can be effectively handled using multi-reference methods that incorporate the so-called “double-*d* shell effect” [30,31] (that is, a 3$d^{\prime }$-shell has to be included in the active space, which is essentially the 4*d*-shell while allowing for mixing with 4*p* when symmetry permits). Additionally, in cases where the Hartree–Fock (HF) reference wavefunctions are qualitatively correct, single-reference high-order coupled-cluster methods are also applicable. Given that both copper and iron compounds share similar sources of error, it is reasonable to anticipate comparable performance across various functionals applied to these molecules.

CuF has been studied using microwave rotational spectroscopy. From the experimental NQCC value of eQq = 21.9562 MHz for ^63^CuF [32] and NMQ value of *Q* = −0.220 barn for ^63^Cu [33], the experimental principal value $V_{c}$ of EFG is determined to be −0.425 a.u. using Equation (Equation 1). Appendix A in the Appendix A compiles the theoretical $V_{c}$ results for the Cu nucleus in CuF ($r_{e}$ = 1.7449 Å [32]) computed by hybrid functionals, double-hybrid functionals, and advanced ab initio methods. As seen in the table, $V_{c}$ = −0.449 a.u. by CCSD(T) closely aligns with the experimental value, and this result improves to −0.433 a.u. when accounting for the correlations from the innermost core electrons. Further enhancement of the results necessitates the corrections from the complete basis set limit and high-order excitations beyond perturbative triples [26]. The EFG results obtained from three multi-configurational methods utilizing nine active orbitals (corresponding to Cu 3*d*4*s* and F 2*p*) are significantly inaccurate. However, after incorporating the Cu 3$d^{\prime }$-shell into the active space, leading to a total of fourteen active orbitals, the new EFG results are qualitatively accurate. Notably, the MRAQCC result of −0.367 a.u. is only 0.058 a.u. larger than the experimental value. To elucidate the discrepancies in the results, we conduct a natural bond orbital (NBO) analysis [34] on the canonical or natural orbitals produced by the aforementioned methods, with the natural electron configurations [35] of Cu detailed in Appendix A. For the three types of reference wavefunctions, the copper atom in the CuF molecule demonstrates an occupation of approximately 3$d^{10}$4$s^{0}$ as predicted by HF and CASSCF(14*o*), while CASSCF(9*o*) supports the 3$d^{9}$4$s^{1}$ occupation. It is evident that CASSCF(9*o*) markedly overestimates the energy associated with the 3$d^{10}$ occupation due to a substantial correlation energy loss. Therefore, the strong correlations observed in CuF arise from the competition among different 3*d* orbital occupations, rather than from the 3d−4*d* correlations (CASSCF(14*o*) has contained all the 3d→4d excitations but its $V_{c}$ result is still not satisfactory), which is likely applicable to other compounds involving 3*d* metals.

In Appendix A, only half of the hybrid functionals and half of the double-hybrid functionals exhibit relatively small errors within 0.22 a.u. Additionally, the double-hybrid functional DSD-PBEP86 also performed well in our previous study involving seventeen iron compounds [3], and therefore eleven hybrid functionals and fourteen double-hybrid functionals are initially selected for further analysis. Appendix A also details the natural electron configurations of copper determined by two well-performing functionals (BH&HLYP and r2SCAN-CIDH) and two less effective functionals (B3LYP and B2PLYP), revealing that these configurations are quite similar irrespective of the EFG results. Therefore, it can be concluded that the EFG error in DFT calculations primarily originates from the inner electrons, which, however, are not addressed in the NBO analysis. Since the correlations in core orbitals are relatively weak, the core orbitals at the HF level may be regarded as nearly exact and can be used to cure the poor functionals. For instance, when examining the B3LYP functional, substituting each of its core orbitals with the corresponding HF orbital reveals a significant change in the $V_{c}$ value for the three Cu 3*p*-like orbitals, from the original 0.350 a.u. to −0.375 a.u. Moreover, increasing the proportion of the HF term in the functional recipe yields a similar effect [17], as evidenced by the hybrid functional results presented in Appendix A, including those from the PBE*n* and TPSS*n* families. Therefore, the $V_{c}$ error associated with the copper nucleus is influenced not only by the treatment of valence electrons but also by the inner 3*p* electrons. This conclusion aligns with findings from the study of CuCl as well [19], and has been attributed to the defects in the exchange part of the functional as noted by Schwerdtfeger et al. [16,17].

The second testing system examined is [Fe=O(tmc)(NCCH_3_)]^2^+ (i.e., the molecule **24** in the test set), which exhibits an experimental NQS value of $\Delta E_{Q}$ = −1.24 mm/s for the ^57^Fe nuclide [36]. Our previous research indicates that its $\Delta E_{Q}$ is difficult to predict using B3LYP [3], and the complete results by different functionals have been provided in the next subsection. Following initial screening, all eleven hybrid functionals demonstrate satisfactory performance, and the predicted $\Delta E_{Q}$ values of ^57^Fe are close to the experimental one. For the double-hybrid functionals, however, two of them yield inaccurate predictions: 2.19 mm/s by $r^{2}$SCAN-QIDH and 2.39 mm/s by P$r^{2}$SCAN69. Consequently, these two functionals have to be excluded from further calculations.

In conclusion, our formal evaluation includes a total of eleven hybrid functionals and twelve double-hybrid functionals (cf. Appendix A for the name list).

### 2.2. Comparison of Density Functionals

The thirty-two medium-sized iron compounds are illustrated in Figure 1, and their optimized Cartesian coordinates are taken from the literature [36,37,38,39]. Considering the limitations of the computational efficiency of double-hybrid functionals, only the molecules containing approximately 70 atoms or fewer are selected, with the largest being Fe(PyO)I(ArS)(CO)_2_PPh_3_ (**19**) which comprises 71 atoms. Notably, the ferrocene molecule [Fe(C_5_H_5_)_2_; **26**] has two distinct conformations: the eclipsed global minimum in $D_{5h}$ symmetry and the staggered saddle point in $D_{5d}$ symmetry with a tiny energy difference between them [40,41]. Our tests indicate that there is a negligible difference in $\Delta E_{Q}$ between these two conformations, and therefore the coordinates of the staggered conformation from Reference [38] are adopted.

Figure 2 shows the error distribution of theoretical $\Delta E_{Q}$ values for the selected 32 iron compounds, and the raw data along with experimental $\Delta E_{Q}$ values are provided in Appendix A for eleven hybrid functionals and Appendix A for twelve double-hybrid functionals, respectively, in the Appendix A. The signs of the experimental $\Delta E_{Q}$ values are taken from the literature when available, including the experimental measurements for [Fe(H_2_O)_5_NO]^2^+ (molecule **4**) [42], [FeS_4_C_8_O_4_]^2^− (**7**) [43], [Fe(SPh)_4_]^2^− (**10**) [43], and the molecules collected in References [36,39]. In cases where experimental data were not identified, the signs are determined based on the theoretical results from this study. As discussed in Section 4.1, predicting the signs of $\Delta E_{Q}$(^57^Fe) in theoretical calculations can be difficult when $|\Delta E_{Q}|<$ 0.4 mm/s or η>3/4; these specific iron compounds are indicated with an asterisk in the tables.

In contrast to the calculations of nuclear contact density, which remains largely unaffected by the choice of functionals [13], the values of $\Delta E_{Q}$ are significantly influenced by different functionals. As seen in Appendix A, all the selected hybrid and double-hybrid functionals generally provide reasonable predictions of $\Delta E_{Q}$ values for most molecules. However, for the remaining molecules, the $\Delta E_{Q}$ results exhibit considerable variation. Notably, the case of [Fe(por)(O_2_)]^−^ (molecule **15**) stands out, as all functionals tend to overestimate its $\Delta E_{Q}$ by 1.5 mm/s or more. It is widely known that the metal–porphyrin systems are characterized by strong static correlations. In the literature, accurate calculations of the iron porphyrin systems can be achieved using full-CI quantum Monte Carlo (FCIQMC) [44,45], generalized active space self-consistent field (GASSCF) [46], density matrix renormalization group (DMRG) [47], and selected-CI (sCI) [48,49] algorithms that can handle very large active spaces. Given that this study is limited to the DFT method, [Fe(por)(O_2_)]^−^ has to be excluded from the error analysis.

Based on the maximum error (MaxE) and mean absolute error (MAE), the leading three hybrid functionals are PBE50 ≺ TPSS38 ≺ *r*SCAN38, with MaxE and MAE being within 1.60 mm/s and 0.31 mm/s, respectively. The top three double-hybrid functionals are $r^{2}$SCAN-CIDH ≺ $r^{2}$SCAN-0DH ≺ PBE-0DH, which exhibit slightly smaller errors. Table 1 provides a comprehensive overview of the results for these six hybrid and double-hybrid functionals. Notably, PBE-0DH achieves the best performance with an MAE of 0.20 mm/s, while PBE50 ranks lowest with an MAE of 0.31 mm/s. The remaining four functionals fall within a middle range, exhibiting MAE values between 0.23 and 0.27 mm/s. When analyzing the absolute values of $\Delta E_{Q}$, as commonly reported in the literature, the MAE values of the majority of the double-hybrid functionals listed in Appendix A, including the top three performers, remain unchanged. Conversely, among the hybrid functionals, only the MAE of *r*SCAN38 is unaffected, while the MAE values of all other hybrid functionals show a decrease. Especially, the MAE of SCAN38 is halved, allowing it to surpass PBE50 and secure the position of the third-ranked hybrid functional. This suggests that most double-hybrid functionals excel in qualitatively predicting the signs of $\Delta E_{Q}$ compared to the hybrid functionals, with the exception of *r*SCAN38. Interestingly, the performance of many double-hybrid functionals is on par with or even inferior to that of the hybrid functional *r*SCAN38, highlighting significant opportunities for further optimization in the development of double-hybrid functionals.

## 3. Discussion

Among the six functionals presented in Table 1, the most significant discrepancies with MaxE exceeding 1.0 mm/s are observed in three molecules (excluding molecule **15**), namely [Fe(H_2_O)_5_NO]^2^+ (molecule **4**), Fe(PyO)I(ArS)(CO)_2_PPh_3_ (**19**), and ferrocene (**26**). The MaxE value for molecule **19** arises from an incorrect sign of $\Delta E_{Q}$, as previously discussed, which occurs only in the hybrid functionals PBE50 and TPSS38, whereas the central radical FeNO^2^+ in **4**, according to the DMRG calculations by Boguslawski et al. [50], is a difficult system with strong static correlations, which should be also true for molecule **4**. Regarding ferrocene, while it is typically classified as a single-reference system, its ground state exhibits significant correlation effects [51]. In our results, all hybrid functionals and the truncated hybrid components of double-hybrid functionals consistently overestimate its $\Delta E_{Q}$ by more than 1.0 mm/s. Only the TPSSh hybrid functional that incorporates a small fraction of exact exchange ($a_{x}$ = 0.1) and the pure functional TPSS are able to accurately calculate its $\Delta E_{Q}$, being 2.69 and 2.36 mm/s, respectively. It is evident that the inclusion of the exact exchange term contributes to the deterioration of these results. A closer examination through NBO analysis indicates that the iron atom in ferrocene exhibits the configuration 3$d^{7.1}$4$s^{0.2}$ according to HF calculations, while TPSS suggests 3$d^{7.7}$4$s^{0.2}$ (that is, Fe^0^ with two 4*s* electrons back donated to 3*d*). So, this is the competition between the 3$d^{n-1}$ and 3$d^{n}$ occupations (*n* = 8 now), as seen in the case of CuF. However, perhaps due to the fact that iron possesses fewer 3*d* electrons compared to copper, HF incorrectly predict the 3$d^{n-1}$ occupation for ferrocene. Ideally, the performance of the exchange functional should be consistent with the exact exchange limit by HF. However, our calculations reveal that the TPSS exchange-only functional by removing the correlation functional continues to support the 3$d^{7.7}$4$s^{0.2}$ configuration, suggesting that this configuration is caused by the exchange functional instead of the correlation one. This implies that the favorable $\Delta E_{Q}$ results seen in TPSS and TPSSh arise from a coincidentally correct 3*d* occupation, which is a consequence of inherent limitations in the approximate exchange functional. Similar trends have also been observed in other hybrid functional families, such as PBE*n*, and the truncated hybrid components of double-hybrid functionals. Upon incorporating correlations from virtual orbitals through second-order perturbation (PT2) corrections, the $\Delta E_{Q}$ values for all double-hybrid functionals significantly decrease, with most errors falling below 0.5 mm/s (refer to Appendix A). So, it is clear that ferrocene is distinctly marked by strong static correlations, but it also demonstrates specific dynamic correlation behaviors since it can be partially corrected through the PT2 corrections.

In double-hybrid functional calculations, three types of density are identified: the self-consistent density produced by the truncated hybrid functional (HFun), the unrelaxed PT2 density with fixed molecular orbitals (UnRlx), and the relaxed PT2 density that incorporates orbital response (Rlx) by solving the coupled-perturbed equations. Appendix A presents the $\Delta E_{Q}$ results and associated errors for all three density types, which can assist in identifying the sources of these errors [13]. It is important to highlight that, for the top three double-hybrid functionals PBE-0DH, $r^{2}$SCAN-0DH, and $r^{2}$SCAN-CIDH, their HFun and UnRlx densities exhibit the most favorable $\Delta E_{Q}$ results, which not only surpass those of all other double-hybrid functionals but also demonstrate errors that are comparable to those associated with the PBE50 functional. In most instances, the HFun and UnRlx densities yield qualitatively accurate $\Delta E_{Q}$ results, while the Rlx densities provide additional minor corrections. However, for molecules **1**, **6**, **13**, **16**, and **29**, it is evident that the Rlx densities from certain functionals can overcorrect $\Delta E_{Q}$, resulting in significantly poorer outcomes. This observation suggests that the proportion of PT2 term is the primary source of error for these molecules. For molecules exhibiting strong static correlations, such as **4** and **15**, the HFun and UnRlx densities are already qualitatively inaccurate, rendering any corrections from the Rlx densities ineffective.

The discrepancies in the signs of $\Delta E_{Q}$ presented in Appendix A can be attributed to several key factors:The parameter η approaches the critical threshold of 0.75, particularly illustrated by molecule **19**. Notably, the η values predicted by certain standard and truncated hybrid functionals even exceed the critical threshold.An improper proportion of the PT2 term contributes to overcorrection, as evidenced by the results of molecule **29** obtained from several double-hybrid functionals.The self-consistent field (SCF) iterations utilizing some (truncated) hybrid functionals converge to distinct occupation patterns within the Fe 3*d*-shell. For instance, in molecule **7**, the contributions of β electrons in 3*d* to the EFG tensor are minimal in the elements $V_{xz}$ and $V_{zx}$ (depending on the coordinate orientation employed in our calculations); however, these contributions are erroneously calculated as −1.2 a.u. by SCAN38, SCAN50, and $r^{2}$SCAN38, leading to an incorrect sign reversal upon diagonalization. A comparable case is also observed in molecule **25**, where the contributions of β electrons in 3*d* to $V_{yz}$ and $V_{zy}$ are −1.0 a.u. but are significantly underestimated to be 0.1 a.u. by the truncated hybrid functional components in $r^{2}$SCAN0-2 and DSD-PBEP86.

There may be other reasons, such as SCF converging to the configurations with varying 3*d* occupations, a phenomenon observed in CuF and ferrocene, but in our testing we have not encountered any instances where this leads to a reversal in the sign of $\Delta E_{Q}$(^57^Fe).

Based on the aforementioned findings, it can be concluded that there is no universally applicable functional for EFG and Mössbauer NQS calculations of the selected iron compounds. Nevertheless, by excluding the most tricky molecule **15** characterized by strong static correlations, some valuable insights can still be obtained. Among the various hybrid and double-hybrid functionals evaluated, PBE-0DH stands out as the most effective, and the calculated $\Delta E_{Q}$ parameters for the tested compounds show good agreement with the experimental ones with an MAE of only 0.20 mm/s. Among the less optimal functionals, *r*SCAN38, $r^{2}$SCAN-CIDH, and $r^{2}$SCAN-0DH demonstrate comparable performance, with MAE values ranging from 0.23 to 0.25 mm/s. However, the hybrid functional *r*SCAN38 shows greater promise since it is superior to the two double-hybrid functionals $r^{2}$SCAN-CIDH and $r^{2}$SCAN-0DH in computational efficiency and has the ability to predict the sign of $\Delta E_{Q}$. Unfortunately, these newly developed functionals, along with the RIJCOSX approximation, have not been implemented in most quantum chemistry programs. Therefore, if a slight reduction in the tolerance for $\Delta E_{Q}$ accuracy is acceptable and the signs of $\Delta E_{Q}$ are not critical, the hybrid functionals BH&HLYP and M06-2X are both good options for predicting satisfactory $|\Delta E_{Q}|$ results in most cases.

In the literature over the past decade, numerous studies have explored $\Delta E_{Q}$(^57^Fe) using various functionals [13,52,53,54]. However, readers may notice that the conclusions drawn in these studies differ from those presented in our research. For instance, some studies suggest that pure functionals outperform hybrid functionals (our results show it is correct only in some cases), or double-hybrid functionals do not perform as well as hybrid ones. The reasons for the varied conclusions can be attributed to several factors, including the following:1.The exclusion of scalar relativistic effects in DFT calculations.2.The electronic correlations present in the dataset, which may be relatively straightforward to manage or distinctly unique.3.The systematic exclusion of pure functionals alongside the inclusion of newly developed hybrid and, particularly, double-hybrid functionals in the evaluation.4.The optimized basis sets for contact density calculations may be inadequate for EFG.5.The neglect of the sign of $\Delta E_{Q}$.6.The SCF iterations converge towards a specific excited state.7.The molecular structures optimized by different methods may affect the errors, among other factors.

Some of these factors may also offset one another, resulting in improved outcomes. Consequently, it is advisable to consider these aspects in future researches, and especially the “Two Wrongs Make a Right” phenomenon in terms of functionals: the incorrect behavior of electronic density predicted by the exchange functional in dealing with transition metals [16,17] might surprisingly result in the correct electronic configuration of Fe whereas the HF method yields the contrary, as evidenced in our results of ferrocene (**26**). Moreover, the effectiveness of the implicit solvent model in simulating the molecular crystal environment has not been comprehensively assessed in the existing literature and in this study, while it often demonstrates superior performance compared to gas phase calculations [36,52], but lacks essential information regarding crystal packing and anisotropic interactions. Alternative approaches for modeling solid environments are worth trying, including the explicit solvent model combined with QM/MM [55], embedded many-body expansion [56], and the more precise (relativistic) DFT method with periodic boundary conditions [57,58], which can significantly enhance the description of the lattice environment and thereby improve the accuracy of theoretical simulations.

## 4. Materials and Methods

### 4.1. Electric Field Gradients and Nuclear Quadrupole Interactions

Any nucleus with a nuclear spin I>1/2 exhibits a non-zero NQM tensor Q, in addition to a scalar NQM value *Q* as collected in the literature [33,59]. The EFG tensor V provides insight into the asymmetric charge distribution surrounding the nucleus. Both Q and V are symmetric and traceless 3 × 3 matrices. The interactions between Q and V can lead to various NQI quantities that can be measured experimentally. Therefore, a primary objective of theoretical research is to calculate EFG as accurately as possible.

By convention, V is transformed into $V^{\prime }$ in a special principal axis system, resulting in $V^{\prime }$ being represented as a diagonal matrix with the eigenvalues $|V_{a}\left|⩽\right|V_{b}\left|⩽\right|V_{c}|$. Because of its traceless nature (i.e., $V_{a}+V_{b}+V_{c}=0$), the EFG tensor can be described using just two coordinate-independent parameters: the principal value $V_{c}$ and the asymmetry parameter $\eta =|\left(V_{a}-V_{b}\right)/V_{c}|$ [1]. When η=0 (i.e., $V_{a}$ = $V_{b}$ = $-V_{c}/2$), the nucleus is positioned at the center of an axisymmetric molecule along the $C_{n}$ axis (n> 2); in other words, there is only one independent parameter $V_{c}$ in this special case. Examples include the linear molecule HCN (for all the three nuclei), PCl_5_ in $D_{3h}$ symmetry (P nucleus), and Ge(CN)_4_ in $D_{4h}$ symmetry (Ge nucleus). It is important to highlight that, for molecules exhibiting spatial degeneracy (e.g., FeO has a doubly degenerate ground state $X^{5}\Delta$), it is essential to compute the sum of the V tensors for all the degenerate components, and failing to do so may lead to a breakdown of symmetry.

After $V_{c}$ being available, NQCC (referred to as eQq) can be calculated by the definition(1)$$\begin{array}{ccc} & & eQq\;\left(\mathrm{in}\mathrm{MHz}\right)=C\;Q\;V_{c}\;, \end{array}$$(2)$$\begin{array}{ccc} & & C=2\times 10^{-28}\;R_{\infty }\;a_{0}^{-2}\;c=234.96478\;, \end{array}$$
where *Q* and $V_{c}$ are expressed in barns (1 barn = $10^{-28}$
$m^{2}$) and atomic units (a.u.), respectively, C is a conversion factor, $R_{\infty }$ is the Rydberg constant (2$R_{\infty }$ = 1 Hartree = 219,474.63137 ${cm}^{-1}$), $a_{0}$ = 0.5291772083 Å is the Bohr radius, and $c\approx 3\times 10^{10}$ cm/s is the speed of light in the CGS unit system (cf. the notation of Equation (47) in Reference [60]).

Another important NQI quantity is the Mössbauer NQS (represented by the symbol $\Delta E_{Q}$), which also depends on η and *I*. The electric quadrupole interaction energy associated with the magnetic quantum number $m_{I}$ ($m_{I}$ = −I, −I+1, *…*, *I*) is [1](3)$$\begin{array}{c} E_{Q}\left(m_{I}\right)=\frac{eQq}{4I(2I-1)}\left[3{m_{I}}^{2}-I\left(I+1\right)\right]{\left(1+\frac{\eta^{2}}{3}\right)}^{1/2}\;. \end{array}$$
For the first excited state of ^57^Fe nuclide, *I* is 3/2; thus, the energy difference between $E_{Q}\left(\pm 3/2\right)$ and $E_{Q}\left(\pm 1/2\right)$ is(4)$$\begin{array}{ccc} & & \Delta E_{Q}\;\left(\mathrm{in}\mathrm{mm}/s\right)=F\;\frac{Q\;V_{c}}{2}{\left(1+\frac{\eta^{2}}{3}\right)}^{1/2}\;, \end{array}$$(5)$$\begin{array}{ccc} & & F=C\frac{c}{E_{\gamma }}=20.21237\;, \end{array}$$
where *Q* is 0.160 barn for ^57^Fe in the *I* = 3/2 excited state [33], $E_{\gamma }$ = 34.84924 × $10^{11}$ MHz (=14.413 KeV [1]) is the γ-radiation energy of ^57^Fe thus $E_{\gamma }/c$ = 11.6248 MHz·s/mm, and the meanings of the other symbols have been clarified previously.

Due to the difficulty in measuring the sign of $\Delta E_{Q}$ experimentally and the frequent inconsistent signs of $\Delta E_{Q}$ between theoretical calculations and experimental measurements, only the absolute values of $\Delta E_{Q}$ are usually compared in the literature. However, in the context of NQCC investigation, the signs of eQq are crucial (see the examples of CuX [16,17,18,19,20]), implying that it is not appropriate to take the absolute values of $\Delta E_{Q}$ in a general way [52]. Furthermore, the intricate electronic structures of iron compounds mean that overlooking the signs of $\Delta E_{Q}$ could obscure fundamental shortcomings of theoretical approaches. There are only two exceptions where determining the sign of $\Delta E_{Q}$ becomes challenging, even at advanced theoretical levels, which consequently diminishes its physical meaning.

If $V_{c}$ is very small or even zero (for example, the sulfur nucleus in SF_6_ with $O_{h}$ symmetry), some minor theoretical errors may result in a swap between $V_{b}$ and $V_{c}$, so the absolute value of the new $V_{c}$ (i.e., the old $V_{b}$) remains nearly unchanged but with an opposite sign. The suggested effective range for $V_{c}$ is $|V_{c}|$ > 0.25 a.u., which approximately corresponds to $|\Delta E_{Q}|$ > 0.4 mm/s for ^57^Fe, as indicated by Equation (Equation 4).η approaches one, i.e., $V_{a}\approx$ 0 and $V_{b}\approx -V_{c}$, which leads to an uncertainty regarding the sign of $V_{c}$ since $V_{b}$ and $V_{c}$ may be interchanged by theoretical errors (cf. page 95 of Reference [1]). The schematic structure for the case of η= 1 is the Ge nucleus in the model molecule “GeHe_2_F_4_” with $C_{2v}$ symmetry, as illustrated in Figure 3. In this model system, the central Ge nucleus experiences a symmetrical charge distribution along the *z*-axis while the *x*- and *y*-directions show asymmetric charge distributions of equal magnitude. Consequently, $V_{a}=V_{zz}=$ 0. It has been found in real systems that even a minor adjustment in the dihedral angle can cause a reversal of the sign of $\Delta E_{Q}$ when the η value surpasses a specific critical point [39]. In this work, η<3/4 is suggested to make the sign of $\Delta E_{Q}$ valid (that is, $|V_{a}/V_{c}|>1/8$ or equivalently $|V_{b}/V_{c}|<7/8$).

### 4.2. Hybrid and Double-Hybrid Density Functionals

Hybrid density functionals incorporate an exact exchange term in the HF form. These functionals primarily consist of three-parameter (3P) hybrid functionals [61,62] and one-parameter (1P) hybrid functionals [63], characterized respectively by the following energy formulas:(6)$$\begin{array}{c} E_{H}\left(\rho \right)=\left\{\begin{array}{cc} a_{1}\cdot E_{x}^{\mathrm{HF}}+\left(1-a_{1}\right)E_{x}\left(\rho \right)+a_{2}\cdot E_{x}\left(\nabla \rho \right)+E_{c}\left(\rho \right)+a_{3}\cdot E_{c}\left(\nabla \rho \right) & \left(3P\right)\\ a_{x}\cdot E_{x}^{\mathrm{HF}}+\left(1-a_{x}\right)E_{x}\left(\rho ,\nabla \rho \right)+E_{c}\left(\rho ,\nabla \rho \right) & (1P)\;. \end{array}\right. \end{array}$$

In this study, the following hybrid functionals are selected: BH&HLYP [61], B3LYP [62], CAM-B3LYP [64], PW6B95 [65], M06-2X [66], ωB97XD [67], PBE*n* family (PBE0 [63], PBE38 [68], and PBE50 [69]), TPSS*n* family (TPSSh [70], TPSS0, TPSS38, and TPSS50) [8], SCAN*n* family (SCAN0 [71], SCAN38, and SCAN50) [8], *r*SCAN*n* family (*r*SCAN0, *r*SCAN38, and *r*SCAN50) [9], and $r^{2}$SCAN*n* family ($r^{2}$SCAN0 [72], $r^{2}$SCAN38 [9], and $r^{2}$SCAN50 [72]). This study excludes pure functionals due to their poor performance in computing the electronic structures of 3*d* metal-containing systems, as found in the studies of CuX [16,17,18,19,20].

Double-hybrid density functionals enhance this approach by incorporating a PT2 term, typically without the need for orbital optimization. There are mainly two categories of double-hybrid functionals: the B2PLYP type introduced by Grimme [73] and the XYG3 type developed by Zhang, Xu, and Goddard [74]. Their general energy formulas are [75].(7)$$\begin{array}{c} E_{\mathrm{DH}}\left(\rho \right)=\left\{\begin{array}{cc} {\left[E_{H}\left(\rho \right)-a_{c}\cdot E_{c}\left(\rho \right)\right]}^{\mathrm{SCF}}+{\left[a_{c}\cdot E_{\mathrm{PT}2}\right]}^{\mathrm{non}-\mathrm{SCF}} & \left(B2\mathrm{PLYP}\right)\\ {\left[E_{H}\left(\rho \right)\right]}^{\mathrm{SCF}}+\Delta E_{h}\left(\rho \right)+{\left[a_{c}\cdot E_{\mathrm{PT}2}\right]}^{\mathrm{non}-\mathrm{SCF}} & \left(\mathrm{XYG}3\right) \end{array}\right. \end{array}$$
with $\Delta E_{h}\left(\rho \right)={\left[E_{h}\left(\rho \right)\right]}^{\mathrm{non}-\mathrm{SCF}}-{\left[E_{H}\left(\rho \right)\right]}^{\mathrm{SCF}}$, where subscript “SCF” means orbital optimization through SCF iterations, “H” denotes a full hybrid functional for B2PLYP or a standard hybrid functional (like B3LYP and PBE0) for XYG3, while “h” indicates the hybrid component of the double-hybrid functional in the XYG3 formulation, which is a reparameterized version of the standard hybrid functional.

Since the natural orbital and expectation value algorithms associated with the XYG3 type have not been extensively implemented, this study focuses solely on the B2PLYP type of double-hybrid functionals, including B2PLYP [73], *m*PW2PLYP [76], Martin’s reparameterizations of B2PLYP (B2GP-PLYP, B2K-PLYP, and B2T-PLYP) [77,78], ωB97X-2 [79], DSD-BLYP [80], DSD-PBEP86 [81], PBE-0DH [82] and its range-separated version RSX-0DH [83], PWPB95 [84], DSD-PBEB95 [85], PBE-QIDH [86] and its range-separated version RSX-QIDH [87], ωB2PLYP and ωB2GP-PLYP [88], ωPBEPP86 and ωB88PP86 [7], 2019 version of DSD/DOD family [6] (DOD-SCAN-D3(BJ), noDispSD-SCAN69, revDSD-PBEP86-D3(BJ), revDSD-BLYP-D3(BJ), and revDOD-PBEP86-D3(BJ)), and 2023 version of $r^{2}$SCAN family [89] ($r^{2}$SCAN-0DH, $r^{2}$SCAN-CIDH, $r^{2}$SCAN-QIDH, $r^{2}$SCAN0-2, P$r^{2}$SCAN50, and P$r^{2}$SCAN69). Certain spin-component scaled (SCS) or spin-opposite scaled (SOS) double-hybrid functionals are specifically defined for excitation energy calculations without affecting the ground state results of the parent functionals (for example, SCS/SOS-B2PLYP21 vs. B2PLYP), and, therefore, these functionals are not pertinent to the current study.

### 4.3. Computational Methods

Firstly, both ORCA 6.0 [90,91] and BDF 2024A [92] program packages are utilized to perform unrestricted BH&HLYP [61] calculations for each molecule independently, ensuring consistency in total energy and $\Delta E_{Q}$ values. Scalar relativistic effects are addressed using the spin-free X2C relativistic Hamiltonian [21,22,23,24] along with the Gaussian-type finite-size nuclear charge distribution [93]. Compared with the approximate relativistic Hamiltonians, X2C offers significant advantages in terms of “simplicity, accuracy, and efficiency” [22] and therefore has been utilized in both this study and our previous Mössbauer parameter investigations [2,3]. The integration grid level is set to defgrid3 in ORCA and ultrafine in BDF, respectively. Since Mössbauer spectroscopy is measured in the solid phase, methanol serves as a solvent to implicitly simulate the crystalline environment using the polarizable continuum model (PCM) [94]. This approach, in conjunction with the conductor-like screening solvation model (COSMO), is deemed appropriate for organometallic systems [36,37,38,52,53,95,96,97]. To enhance the efficiency of integral calculations, the RIJCOSX [98] and aMPEC+aCOSX [99] approximations have been enabled in ORCA and BDF, respectively. For the majority of molecules, both programs yield very close total energies and $\Delta E_{Q}$ values, but, for two molecules, [Fe(H_2_O)_5_NO]^2^+ (**4**) and [Fe(PyS)I(CO)_2_PPh_3_] (**20**), BDF produces slightly lower energies. As a result, the utilities molden2fch and fch2mkl in MOKIT 1.2.5 [100] are utilized to transfer molecular orbitals from BDF to ORCA.

Next, the ORCA program package is used to conduct a range of hybrid and double-hybrid functional calculations. To maximize convergence to the same electronic states, initial orbitals are read from the previously performed BH&HLYP calculations. Sample ORCA input files for undefined functionals can be found in the original literature associated with these functionals. The default setting of frozen core in ORCA is adopted in the double-hybrid functional calculations, that is, in addition to valence electrons, the semi-core electrons in 3*s*3*p* of Fe and (n-1)*d* of post-3*d* elements are also correlated.

All the calculations are carried out using the x2c-TZVPPall-f (for pre-3*d* atoms) and x2c-TZVPPall [101] relativistic basis sets. As noted by Santra et al. [13], the standard x2c-TZVPPall basis set of the iron atom performs poorly in the EFG calculations and therefore has to be slightly modified in our study: the functions with angular quantum number l>0 are decontracted to enhance flexibility in describing the electron density distribution near the nucleus; in contrast, the spherical *s*-functions, which do not explicitly contribute to EFG, are retained in their contracted form. Furthermore, EFG calculations often require the use of some very tight functions with l>0 to achieve saturation of the basis set, which is essential for precise EFG results [26]. Our test calculations at the HF level suggest that the x2c-TZVPPall basis set for Fe is deficient in tight *p*- and *d*-functions. To achieve acceptable convergence with the $V_{c}$ error remaining below 0.01 a.u. (or a $\Delta E_{Q}$(^57^Fe) error of less than about 0.016 mm/s; see Appendix A), it is recommended to add one additional tight *p*-function and three tight *d*-functions, and their Gaussian exponents generated by the even-tempered series can be found in Reference [3].

In addition to the 32 iron compounds, the strongly correlated gas-phase molecule CuF is also calculated using a variety of hybrid and double-hybrid density functionals, as well as several ab initio methods implemented in the Molpro 2015.1 program package [102]. The single-reference ab initio methods employed include CCSD (coupled-cluster with single and double excitations) and CCSD(T) (CCSD with perturbative triple excitations) [103], while the multi-configurational methods are CASSCF (complete active space self-consistent field), internally contracted MRCI (multi-reference configuration interaction with single and double excitations) [104], and internally contracted MRAQCC (multi-reference averaged quadratic coupled-cluster with single and double excitations) [105]. The calculation process of EFG at these theoretical levels is detailed in Reference [3]. The scalar relativistic effects are calculated again at the spin-free X2C level, and the core–valence correlations from the Cu 3*s*3*p* electrons are counted in the double-hybrid functional and advanced ab initio calculations. The basis sets employed are x2c-TZVPPall-f for the fluorine atom and x2c-TZVPPall for the copper atom [101], respectively, but the latter requires some modifications, similar to the adjustments made for the iron atom, by supplementing four additional Gaussian exponents with $\alpha_{p}$ = 5813.6650 and $\alpha_{d}$ = 1428.9269, 491.72241, and 169.21155.

## 5. Conclusions

While high-precision EFG calculations for small molecules have been successfully addressed, as demonstrated in References [20,26], predicting the Mössbauer NQS for medium-sized molecules continues to pose significant challenges. In this context, the DFT method is indispensable. Consequently, identifying the “best” density functionals tailored for EFG calculations has emerged as a critical objective in this field.

In this study, the $\Delta E_{Q}$ values of the ^57^Fe nuclide for 32 iron-containing molecules have been calculated using selected hybrid and double-hybrid functionals, and are compared with experimental values. In error statistics, the signs of $\Delta E_{Q}$ with the considering of two exceptional cases, namely $|\Delta E_{Q}|$(^57^Fe) < 0.4 mm/s and η>3/4, have also been examined. Our results lead us to recommend the following functionals for NQS calculations of ^57^Fe nuclide.

The double-hybrid functional PBE-0DH demonstrates strong agreement with experimental results, outperforming other functionals with an MAE of 0.20 mm/s.If computational cost is a primary concern, the hybrid functional *r*SCAN38 is recommended, as it exhibits a slightly larger MAE of 0.25 mm/s, while still delivering satisfactory results for most molecules.In cases where the quantum chemistry program does not support the aforementioned functionals, the older hybrid functionals BH&HLYP and M06-2X can be utilized, albeit with a greater MAE of 0.33 mm/s.

The most challenging systems to study are those characterized by strong static correlations. While individual hybrid, double-hybrid, or even pure functionals can provide reasonable results, they exceed the capabilities of DFT and the results often hinge on chance, showing the complexities involved in accurately modeling such systems. In contrast, advanced multi-configurational methods, such as DMRG and sCI, offer significant advantages as demonstrated in our previous research on Mössbauer parameters [2,3].

The pursuit of identifying optimal functionals for Mössbauer NQS calculations of iron compounds remains an active area of research, which, however, has not garnered significant attention in the realm of functional development. Future advancements in the development of new functionals may incorporate $\Delta E_{Q}$ data into their evaluation, whereas the contact density data are still not enough. This approach may represent a promising avenue for enhancing the accuracy of DFT calculations.

## Figures and Tables

**Figure 1 ijms-26-02821-f001:**
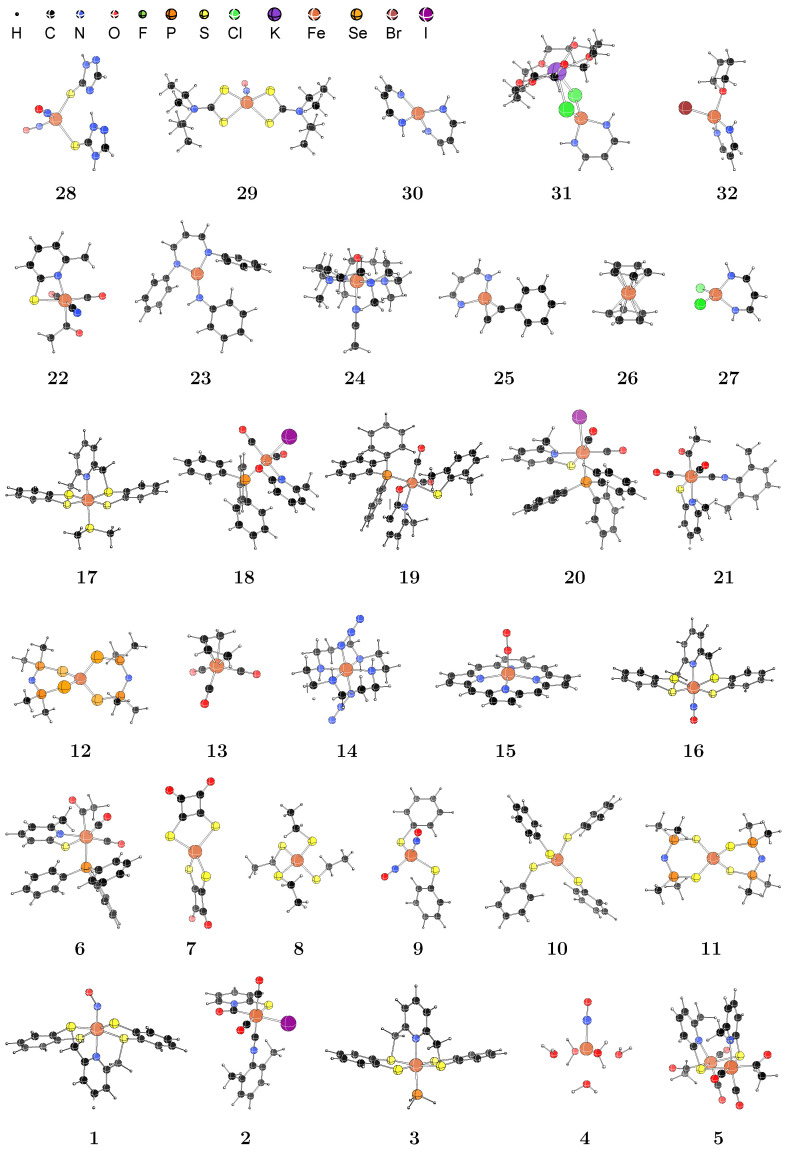
Structures of molecules Fe(NO)(pyS_4_) (**1**), Fe(SC_5_H_4_N-CO)I(CO)_2_(ArNC) (**2**), Fe(PH_3_)(pyS_4_) (**3**), [Fe(H_2_O)_5_NO]^2^+ (**4**), [Fe(PyS)(ac)(CO)_2_]_2_ (**5**), Fe(PyS)(ac)(CO)_2_PPh_3_ (**6**), [FeS_4_C_8_O_4_]^2^− (**7**), [Fe(SEt)_4_]^−^ (**8**), [Fe(SPh)_2_(NO)_2_]^−^ (**9**), [Fe(SPh)_4_]^2^− (**10**), Fe[(SPiPr_2_)_2_N]_2_ (**11**), Fe[(SePiPr_2_)_2_N]_2_ (**12**), Fe(4−butadiene)(CO)_3_ (**13**), *trans*-[Fe(cyclam)(N_3_)_2_]^+^ (**14**), [Fe(por)(O_2_)]^−^ (**15**), [Fe(NO)(pyS_4_)]^+^ (**16**), Fe(SMe_3_)(pyS_4_) (**17**), Fe(PyO)I(CO)_2_PPh_3_ (**18**), Fe(PyO)I(ArS)(CO)_2_PPh_3_ (**19**), Fe(PyS)I(CO)_2_PPh_3_ (**20**), Fe(Pys)(ac)(CO)_2_(ArNC) (**21**), [Fe(PyS)(ac)(CN)(CO)_2_]^−^ (**22**), L^tBu^FeNHtolyl (**23**), [Fe=O(tmc)(NCCH_3_)]^2^+ (**24**), L^tBu^Fe(HCCPh) (**25**), staggered ferrocene (**26**), ^Me^L^Me,Me^FeCl_2_ (**27**), Fe(SC_2_H_3_N_3_)(SC_2_H_2_N_3_)(NO)_2_ (**28**), Fe(NO)(dtci-Pr2${)}_{2}$ (**29**), ^Me^L_2_^Me,Me^Fe (**30**), ^Me^L^Me,Me^FeCl_2_K(18-crown-6) (**31**), and L^Me^FeBr(THF) (**32**).

**Figure 2 ijms-26-02821-f002:**
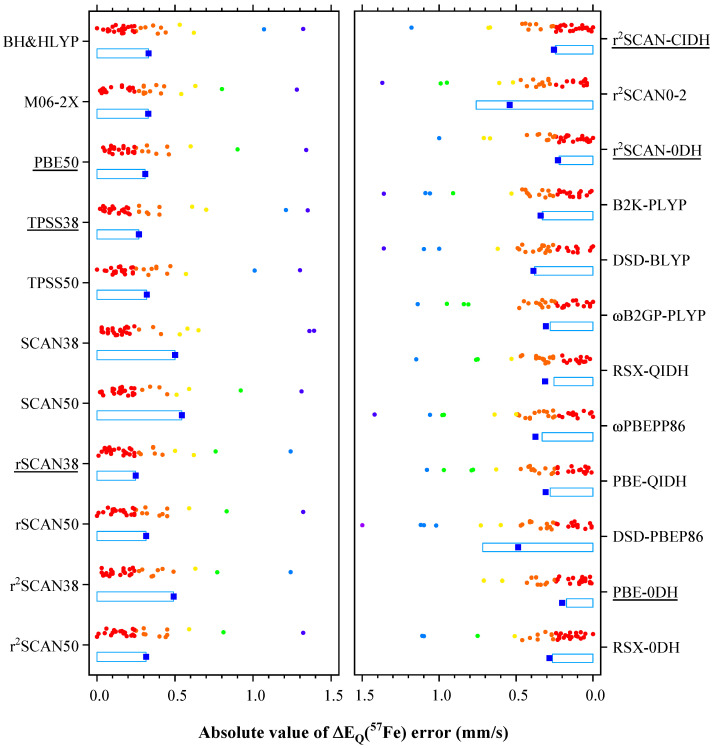
Errors of $\Delta E_{Q}$(^57^Fe) by different hybrid (left panel) and double-hybrid functionals (right panel). Light blue boxes show the ranges of standard deviation, blue squares show the mean absolute errors, and colored dots show the absolute values of errors within 1.5 mm/s for various molecules. The names of top three hybrid and top three double-hybrid functionals are underlined. Molecule **15** has been excluded from the error analysis.

**Figure 3 ijms-26-02821-f003:**
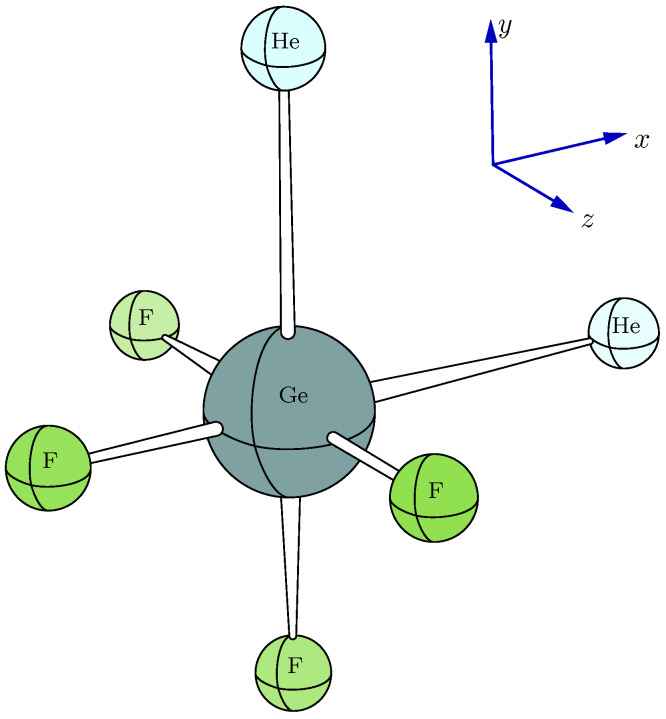
Model system GeHe_2_F_4_ to demonstrate the sign problem of $V_{c}$.

**Table 1 ijms-26-02821-t001:** $\Delta E_{Q}$ results (in mm/s) of ^57^Fe by selected hybrid and double-hybrid ${\mathrm{functionals}}^{\;(a)}$.

${\mathrm{Mol.}}^{\;(b)}$	$N_{\mathrm{atom}}^{2S+1}$	$\eta^{\;(c)}$	H-3	H-4	H-8	DH-1	DH-3	DH-11	Expt.	Ref.
**1** *	$42^{2}$	0.50	−0.32	−0.24	−1.16	−0.44	0.33	−0.32	−0.40	[36]
**2** *	$38^{1}$	0.79	−0.43	−0.38	−0.38	−0.29	−0.29	0.21	0.29	[36]
**3** *	$44^{1}$	0.79	0.74	0.73	0.72	0.74	0.73	0.74	0.69	[36]
**4**	$18^{4}$	0.45	3.00	2.80	2.60	3.28	3.10	2.69	2.10	[37]
**5**	$50^{1}$	0.45	−0.94	−0.91	−0.92	−0.98	−0.95	−0.98	−0.74	[36]
		0.44	−0.94	−0.91	−0.91	−0.97	−0.95	−0.98		
**6**	$59^{1}$	0.34	−1.19	−1.19	−1.20	−1.36	−1.32	−1.39	−1.14	[36]
**7**	$17^{5}$	0.06	−4.08	−4.05	−4.05	−4.06	−4.05	−4.04	−3.97	[37]
**8**	$33^{6}$	0.02	−0.33	−0.35	−0.35	−0.35	−0.36	−0.39	−0.62	[37]
**9**	$29^{2}$	0.26	−0.78	−0.71	−0.78	−0.76	−0.75	−0.67	−0.69	[37]
**10** *	$49^{5}$	0.81	−3.57	−3.51	−3.48	−3.50	−3.48	−3.47	−3.24	[37]
**11**	$43^{5}$	0.11	3.79	3.73	3.73	3.73	3.72	3.71	3.62	[39]
**12**	$43^{5}$	0.12	3.85	3.79	3.80	3.79	3.78	3.77	3.61	[39]
**13**	$17^{1}$	0.12	−1.59	−1.54	−1.56	−1.68	−1.63	−1.64	−1.34	[36]
**14**	$45^{2}$	0.18	−2.01	−1.92	−1.87	−2.03	−1.99	−2.19	−2.24	[36]
**15**	$39^{6}$	0.63	2.52	2.25	2.14	2.33	2.25	2.06	0.62	[36]
**16** *	$42^{1}$	0.76	−1.59	−1.58	−1.58	2.05	1.93	2.06	−1.63	[36]
**17**	$49^{1}$	0.23	0.35	0.35	0.34	0.36	0.36	0.38	0.43	[36]
**18** *	$54^{1}$	0.90	−0.59	−0.53	−0.54	−0.50	−0.50	0.50	0.48	[36]
**19**	$71^{1}$	0.64	0.51	0.52	−0.55	−0.69	−0.67	−0.76	−0.83	[36]
**20** *	$54^{1}$	0.62	0.55	0.49	0.49	0.38	0.39	0.36	−0.35	[36]
**21**	$44^{1}$	0.38	0.97	0.92	0.94	0.92	0.90	0.86	0.73	[36]
**22**	$27^{1}$	0.58	1.13	1.09	1.10	1.10	1.08	1.04	0.89	[36]
**23** *	$44^{5}$	0.98	−1.87	−1.82	−1.84	−1.83	−1.85	−1.82	−1.42	[38]
**24**	$58^{3}$	0.48	−1.40	−1.10	−1.19	−1.37	−1.26	−1.15	−1.24	[36]
**25**	$25^{4}$	0.49	−2.65	−2.66	−2.67	−2.73	−2.72	−2.76	−2.05	[38]
**26**	$21^{1}$	0.00	4.01	3.62	3.65	3.08	3.12	2.78	2.41	[38]
**27**	$13^{6}$	0.42	−0.85	−0.90	−0.87	−0.86	−0.87	−0.92	−1.23	[38]
**28**	$22^{2}$	0.49	−1.29	−1.22	−1.29	−1.27	−1.25	−1.19	−1.12	[37]
**29**	$51^{2}$	0.11	0.79	0.68	0.81	0.43	0.61	0.53	0.89	[37]
**30**	$21^{5}$	0.13	1.89	1.84	1.79	1.79	1.77	1.78	1.80	[38]
**31**	$56^{5}$	0.34	2.56	2.50	2.46	2.54	2.45	2.44	2.10	[38]
**32**	$25^{5}$	0.73	2.68	2.63	2.60	2.60	2.58	2.58	2.36	[38]
${\mathrm{MaxE}}^{\;(d)}$			1.60	1.35	1.24	1.18	1.00	−0.71		
${\mathrm{MAE}}^{\;(d)}$			0.31	0.27	0.25	0.25	0.23	0.20		
${\mathrm{MAE}}^{\;(e)}$			0.28	0.24	0.25	0.25	0.23	0.20		

^(a)^ The hybrid (H) and double-hybrid (DH) functionals are PBE50 (**H-3**), TPSS38 (**H-4**), *r*SCAN38 (**H-8**), *r^2^*SCANCIDH
(**DH-1**), *r^2^*SCAN-0DH (**DH-3**), and PBE-0DH (**DH-11**). See Appendix A for complete data. ^(b)^ The
molecule with an asterisk means that the sign of Δ*E_Q_* is uncertain (see Section 4.1). ^(c)^ Calculated by *r^2^*SCANCIDH.
^(d)^ Maximum error and mean absolute error. Molecule **15** has been excluded. ^(e)^ Mean absolute error of |Δ*E_Q_*|. Molecule **15** has been excluded.

## Data Availability

Data is contained within the article and Appendix A.

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
