# Peer review of "Prediction of 57Fe Mössbauer Nuclear Quadrupole Splittings with Hybrid and Double-Hybrid Density Functionals"

_ijms, 2025, doi:10.3390/ijms26062821_

Round 1
Reviewer 1 Report
Comments and Suggestions for Authors
In this work, the authors studied the performance of eleven hybrid density functionals and twelve double-hybrid density functionals in predicting the NQS values of 57Fe nuclide for 32 iron containing molecules within about 70 atoms.The results are rich and the discussion is thorough. The manuscript can be accepted after a minor revision.
- What do the atoms of different colors in Figure 2 represent? Please have the author mark it.
- In the conclusion section, please ask the author to further condense the obtained results in detail.
- For the calculation method section, please ask the author to cite some relevant literatures.
Author Response
We thank the referees for useful and constructive comments, which we have used to improve the manuscript. In the following we have given the original comments of the referee in black and our replies in the blue font.
Comments 1: What do the atoms of different colors in Figure 2 represent? Please have the author mark it.
Response 1: In Figure 2 (Figure 1 now), we have added element symbols for atoms of different colors.
Comments 2: In the conclusion section, please ask the author to further condense the obtained results in detail.
Response 2: We have reorganized the conclusion section.
Comments 3: For the calculation method section, please ask the author to cite some relevant literatures.
Response 3: In the calculation method section (now the Computational Methods subsection), some literatures about NQS calculation with the implicit solvent model have been cited (Lines 450-451).
Reviewer 2 Report
Comments and Suggestions for Authors
This manuscript explores the ability of different DFT functionals to predict Mössbauer nuclear quadrupole splittings. The work is correctly performed, the used dataset is extensive (unlike in many other similar calibration works) and the conclusions are weel-grounded. There are several possibilities for extenson, most of them outlined also by the autors. It is a nice work and deserves to be published as is.
I have just one additional future sugestion to authors: please consider also the testing of basis set influence on the results with the chosen best functionals – my experience is that the sensitivity of different DFT functionals to the used basus set is sometimes very different, i.e. some functional might need more (or less) extensive basis set than others and it would be nice to have a suggested method for calculations, which is optimised also basis set-wise.
There were also one minor typo, which should be corrected:
1. On page 10, line 352 there is a unit mms/s (should be mm/s).
Author Response
We thank the referees for useful and constructive comments, which we have used to improve the manuscript. In the following we have given the original comments of the referee in black and our replies in the blue font.
Comments 1: I have just one additional future sugestion to authors: please consider also the testing of basis set influence on the results with the chosen best functionals – my experience is that the sensitivity of different DFT functionals to the used basus set is sometimes very different, i.e. some functional might need more (or less) extensive basis set than others and it would be nice to have a suggested method for calculations, which is optimised also basis set-wise.
Response 1: We have added a new Table S5 in Supplementary Materials with the computational results of two molecules, which demonstrates that the currently used basis set is approaching convergence (see line 475 in the main text).
Comments 2: There were also one minor typo, which should be corrected: On page 10, line 352 there is a unit mms/s (should be mm/s).
Response 2: We have made a correction to this typo, located at line 209 of the main text.
Reviewer 3 Report
Comments and Suggestions for Authors
Dear Authors,
To enhance the impact of your manuscript, I respectfully suggest addressing the following points:
1. Previous studies report that pure functionals outperform hybrid functionals for predicting Fe Mössbauer parameters. Your work contradicts these conclusions without explicitly addressing potential reasons for the discrepancy (e.g., differences in basis sets, convergence criteria, or treatment of relativistic effects). A deeper discussion reconciling these conflicting results such as analyzing the role of exact exchange or solvent effects would strengthen transparency and contextualize your findings.
2. The manuscript notes that certain functionals fail to predict the correct sign of NQS for specific molecules. However, the root causes (e.g., exchange-correlation errors, basis set limitations, or inadequate treatment of electron correlation) remain unexplored. A systematic investigation such as comparing spin densities, orbital occupations,could clarify whether these errors arise from methodological shortcomings or electronic structure complexities.
3. While the article states that "none of the functionals produced satisfactory results" for this complex, the underlying reasons (e.g., strong multireference character, spin-state mixing, or ligand-field effects) are not analyzed. Natural orbital analysis or occupation number diagnostics could help determine whether static correlation dominates, necessitating methods beyond DFT (e.g., CASSCF or DMRG).
4. The conclusion designates PBE-0DH as optimal based on error metrics, but a theoretical rationale for its performance (e.g., balance of exact exchange, treatment of dispersion, or compatibility with the PCM model) is lacking. Explicitly linking its success to physicochemical properties (e.g., spin-state energetics or charge transfer) would provide a more robust justification.
5. The use of a polarizable continuum model (PCM) to approximate solid-state effects may inadequately capture explicit crystal packing or anisotropic interactions critical to Mössbauer parameters. A periodic DFT approach or embedded cluster models (QM/MM) could better represent the lattice environment and improve agreement with experiment.
6. While CCSD(T) and multireference methods are briefly mentioned for CuF, extending this comparison across all studied molecules particularly those with open-shell or degenerate states would validate the reliability of DFT and highlight regimes where it remains adequate or inadequate.
Addressing these points would significantly strengthen the manuscript’s scientific depth.
Thank you for considering these suggestions.
Best regards,
Author Response
We thank the referees for useful and constructive comments, which we have used to improve the manuscript. In the following we have given the original comments of the referee in black and our replies in the blue font.
To enhance the impact of your manuscript, I respectfully suggest addressing the following points:
Comments 1: Previous studies report that pure functionals outperform hybrid functionals for predicting Fe Mössbauer parameters. Your work contradicts these conclusions without explicitly addressing potential reasons for the discrepancy (e.g., differences in basis sets, convergence criteria, or treatment of relativistic effects). A deeper discussion reconciling these conflicting results such as analyzing the role of exact exchange or solvent effects would strengthen transparency and contextualize your findings.
Response 1: All the pure GGA functionals have been precluded by the test calculations of CuF according to the calculations by Schwerdtfeger et al, and the same is also true when tested with the second test molecule 24 (they all predict an incorrect sign of NQS which are not provided in the text). Schwerdtfeger et al. pointed out that the exchange functionals have inherent defects when dealing with transition metals. We attribute the opposite conclusion that pure functionals are superior to hybrid functionals to the phenomenon of "Two Wrongs Make a Right," as seen in lines 330-333 of the main text.
Comments 2: The manuscript notes that certain functionals fail to predict the correct sign of NQS for specific molecules. However, the root causes (e.g., exchange-correlation errors, basis set limitations, or inadequate treatment of electron correlation) remain unexplored. A systematic investigation such as comparing spin densities, orbital occupations, could clarify whether these errors arise from methodological shortcomings or electronic structure complexities.
Response 2: In lines 276-295 of the main text, we summarized the various causes of the sign problem.
Comments 3: While the article states that "none of the functionals produced satisfactory results" for this complex, the underlying reasons (e.g., strong multireference character, spin-state mixing, or ligand-field effects) are not analyzed. Natural orbital analysis or occupation number diagnostics could help determine whether static correlation dominates, necessitating methods beyond DFT (e.g., CASSCF or DMRG).
Response 3: This conclusion pertains to the strong multireference character. Apart from the small molecule CuF, these multi-reference systems have all been verified using the DMRG method in the literature. Unfortunately, aside from DMRG calculations, there is no easy-to-use indicator to determine whether a large molecule is a multireference system. The natural configuration of valence electrons by NBO analysis help us identify CuF and ferrocene with the 3d(n-1)-vs.-3d(n) type of multireference character, but it is not helpful for other testing molecules. Truhlar et al. suggested six diagnostic criteria (see the subsection 3.3.2 in J. Phys. Chem. A 2007, 111, 4632), among which the T1 diagnostic by CCSD is the simplest one, which however has exceeded the current computational capabilities when applied to the tested molecules in this article.
Comments 4: The conclusion designates PBE-0DH as optimal based on error metrics, but a theoretical rationale for its performance (e.g., balance of exact exchange, treatment of dispersion, or compatibility with the PCM model) is lacking. Explicitly linking its success to physicochemical properties (e.g., spin-state energetics or charge transfer) would provide a more robust justification.
Response 4: In lines 264-268 of the main text, it is mentioned that its HFun densities of PBE-0DH have the smallest error among all the double-hybrid functionals. Apart from simple systems, however, it is difficult to further analyze the impact of each term in the functional formula (although this is computationally feasible) because these terms can cause the SCF procedure to converge to different electronic configurations or states.
Comments 5: The use of a polarizable continuum model (PCM) to approximate solid-state effects may inadequately capture explicit crystal packing or anisotropic interactions critical to Mössbauer parameters. A periodic DFT approach or embedded cluster models (QM/MM) could better represent the lattice environment and improve agreement with experiment.
Response 5: We agree with the reviewer's viewpoint and have made additions to the paper in lines 335-342.
Comments 6: While CCSD(T) and multireference methods are briefly mentioned for CuF, extending this comparison across all studied molecules particularly those with open-shell or degenerate states would validate the reliability of DFT and highlight regimes where it remains adequate or inadequate.
Response 6: It is well known that CCSD(T) and various multireference methods can only calculate small molecules, while the DLPNO-CCSD(T) approximation currently does not have natural orbitals for property calculations, and its convergence accuracy does not meet the requirements of EFG calculations by numerical differences (about 1.0e-10). The only feasible methods are the large-scale DMRG and sCI methods with acceleration of integration, as mentioned in Conclusions section of the paper.
Addressing these points would significantly strengthen the manuscript’s scientific depth.
Thank you for considering these suggestions.
Round 2
Reviewer 3 Report
Comments and Suggestions for Authors
Dear Authors,
Thank you for your thorough and careful revisions.
Wishing you all the best in your future endeavors.
Best regards,